# Intercoupling of Cascaded Metasurfaces for Broadband Spectral Scalability

**DOI:** 10.3390/ma16052013

**Published:** 2023-02-28

**Authors:** Shaolin Zhou, Liang Liu, Qinling Deng, Shaowei Liao, Quan Xue, Mansun Chan

**Affiliations:** 1School of Microelectronics, South China University of Technology, Guangzhou 510640, China; 2ACCESS—AI Chip Center for Emerging Smart Systems, Hong Kong Science Park, Hong Kong, China; 3Department of Electronic and Computer Engineering, The Hong Kong University of Science and Technology, Hong Kong, China

**Keywords:** metasurface, broadband spectral analysis, cascade control, band-stop filters

## Abstract

Electromagnetic metasurfaces have been intensively used as ultra-compact and easy-to-integrate platforms for versatile wave manipulations from optical to terahertz (THz) and millimeter wave (MMW) ranges. In this paper, the less investigated effects of the interlayer coupling of multiple metasurfaces cascaded in parallel are intensively exploited and leveraged for scalable broadband spectral regulations. The hybridized resonant modes of cascaded metasurfaces with interlayer couplings are well interpreted and simply modeled by the transmission line lumped equivalent circuits, which are used in return to guide the design of the tunable spectral response. In particular, the interlayer gaps and other parameters of double or triple metasurfaces are deliberately leveraged to tune the inter-couplings for as-required spectral properties, i.e., the bandwidth scaling and central frequency shift. As a proof of concept, the scalable broadband transmissive spectra are demonstrated in the millimeter wave (MMW) range by cascading multilayers of metasurfaces sandwiched together in parallel with low-loss dielectrics (Rogers 3003). Finally, both the numerical and experimental results confirm the effectiveness of our cascaded model of multiple metasurfaces for broadband spectral tuning from a narrow band centered at 50 GHz to a broadened range of 40~55 GHz with ideal side steepness, respectively.

## 1. Introduction

In the past decades, electromagnetic (EM) metamaterials (MMs) constructed by periodic arrays of sub-wavelength elements have been acting as exotic platforms for wave-matter interactions due to their anomalous features such as negative permeability [1], negative dielectric constant [2], and negative refractive index [3]. Recently, the planar or two-dimensional (2D) versions of MMs [4], namely metasurfaces [5,6,7,8], further show unprecedented versatility as their bulk counterpart, but with only the deep sub-wavelength thickness that facilitates the application of ultra-compact integration. As they are different from the conventional bulk MMs, metasurfaces intrinsically achieve the amplitude or phase modulations by tailoring individual meta-atoms that are customized and distributed locally in a pixel-wise manner. In this trend, a plethora of metasurfaces-based devices have been demonstrated from the visible to terahertz (THz) and millimeter waves (MMW) regimes for passive and active regulations of the amplitude and phase spectra [9], such as frequency selection [10,11,12], perfect absorption [7,13,14], THz modulators [15,16], electromagnetic-induced transparency [17,18,19], broadband transmitters [20], and detectors [21], beam steering [9], flat lensing [5,22], optical or electromagnetic activity and vortex [23,24,25,26,27] devices, etc. Further, a few advanced features or critical appeals emerge, such as active re-configurability, multi-band spectral control [13,28,29,30], omni-directional independence [31,32], and polarization independence with broadband spectral control [33,34]. Among them, the broadband spectral behaviors with scalability are especially desirable in myriads of THz and MMW applications such as filtering, sensing, radiation, absorption, and so on. 

To date, although diverse schemes have been reported only for the purpose of broad spectral regulations, e.g., by using the composite meta-atoms composed of multi-patches [35], multi-spirals [36], or multi-rings [37], by the cascading of multi-layer surfaces [38,39], by using cone- [34], tapered- [40] or pyramid [41]-shaped atoms, by direct dispersion engineering [7,42,43,44], and by multi-interference and diffraction effect [45], etc., the broadband regulations with scalability are somehow considered less often. Even for those schemes involving the cascading of multiple surfaces in parallel [38,39,46], the effect of interlayer coupling involved in between multiple layers is rarely treated and leveraged for purposeful spectral regulations. In particular, different from the modes coupling or overlapping (which can be termed as intra-layer coupling in our context) for bandwidth enhancement in composite atoms of a single metasurface [47], the interlayer coupling of multiple metasurfaces is treated, in this paper, for broadband spectral tuning with a higher degree of freedom. For such an effect of interlayer coupling on resonance splitting, the Lagrangian method was used early to rigorously treat the near-field interactions of atom resonators, and the MM spectral response of superlattice structures can be tuned by varying the coupling parameters [48]. Recently, the coupled-mode theory is used to simplify the physical understanding of the interlayer couplings of cascaded meta-atoms with distinct types for tunable phase responses [49,50]. 

However, in this paper, to simply provide a concise and feasible route to understand the interlayer couplings of cascaded metasurfaces with similar (or the same) geometries, we tend to derive a simplified model of equivalent circuits of lumped parameters to provide an intuitive guidance for intended broadband spectral tuning. As a proof of concept, we explore the broadband scalability of the band-stop transmission spectra based on cascaded metasurfaces in the MMW regime in this paper. The multi-layer cascaded models are first constructed using a simple composite meta-atom composed of Jerusalem crosses and the complementary resonant rings for the single layer. The interlayer couplings among the cascaded layers are fully interpreted and employed for broadband control with spectral scalability. The hybridized or overlapped resonance modes due to inter-couplings are intuitively modeled by the transmission line (TL) equivalent circuits, which also act as the guidance for scalable and broadband spectral control. Based on the TL analysis for a single meta-surface with only narrow-band behaviors [51], the cascaded models of double and triple metasurfaces with intercoupling are obtained for the scalable tuning of broadband spectral behaviors. As a result, the narrow band of a single metasurface at around 50 GHz is readily expanded to be to around 45 GHz~55 GHz with an ideal sideband steepness. As indicated by the simplified TL equivalent circuit models, the broadband spectral behaviors can be tuned by taking advantage of the relationships between the geometrical scaling and the spectral scaling for hybridized resonance modes splitting or coupling. As a result, our work may provide a promising approach for metasurface-based spectral tuning with broadband scalability for applications in the MMs, THz, and optical regimes, such as the tunable, antireflective coatings, configurable band-pass, or band-stop filters, etc. 

## 2. Methods and Analysis Models

### 2.1. Analysis Model of Single-Layer Metasurface

To probe the inter-coupling effect between multiple metasurfaces cascaded in parallel, we start with the accurate modeling of a single metasurface. First, one type of metasurface is constructed by using the Jerusalem cross inside a square ring as the unit meta-atom cell (Figure 1a). Typically, metasurfaces with such simple capacitive cells of metal elements act as frequency selective devices with transmissive band-stop spectra, as shown in Figure 1b, calculated by the finite element method (FEM). 

Constructed with symmetrically and periodically arranged elements, such a metasurface shows an ideal geometrical isotropy and polarization independence. So, two basic lumped equivalent circuits with similar band-stop transmission spectra can be used for simple modeling: the parallel and series configurations of *LC* circuits, as shown in Figure 2, according to the transmission line (TL) theory [51]. 

Essentially, these two circuits differ in the imaginary parts of their characteristic impedance. Namely, the dominant reactances of the parallel and series LC circuit are $-j\omega /\left(\omega^{2}-\omega_{0}^{2}\right)c$ and $jL\left(\omega^{2}-\omega_{0}^{2}\right)/\omega$, respectively, with the same resonant frequency at $\omega_{0}=1/\sqrt{LC}$. Further, according to the transmission spectrum (Figure 1b) and the characteristic wave impedance (Figure 1b) that is obtained by a retrieval procedure based on scattering coefficients [51,52], the series LC configuration in Figure 2a fundamentally fits well in terms of the characteristic impedance, i.e.,$\;jL\left(\omega^{2}-\omega_{0}^{2}\right)/\omega$, which has a zero point at $\omega_{0}$. Therefore, a series LC circuit based on the TL circuit A in Figure 2a is chosen to model the resonance of the single-layer metasurface shown in Figure 3. 

Herein, to take the ohmic loss into account, a resistor *R* is added into the series with inductance *L*_1_, and *Z_o_* is the vacuum wave impedance. *C*_1_ is the total equivalent capacitance determined by the electric resonance of the single-layer metasurface. It can be modeled as the coupling capacitance *C_in_* between the outer rings and inner Jerusalem crosses in parallel with the external coupling capacitance *C_out_* between the outer neighboring rings, namely,
(1)$$C_{1}\approx C_{in}+C_{out}$$
where $C_{in}\approx \pi \varepsilon_{r}\varepsilon_{0}l_{2}/\ln\left(d_{1}/t\right)$ and $C_{out}\approx \pi \varepsilon_{r}\varepsilon_{0}D/ln\left[\left(P-D\right)/t\right]$ are used for good approximations of the geometries [53], and $d_{1}=D-2g-2w_{2}-l_{1}$ denotes the internal gap between the cross and square ring, $\varepsilon_{r}$ is the relative permittivity of dielectric substrate underneath, *t* is thickness of the metal ring, and other geometries, *l*_2_, *l*_1_, *w*_2_, *D*, and *g*, are shown in Figure 1. Additionally, the inductance of *L*_1_ dominated by outer square rings length can be estimated according to spiral inductance as
(2)$$L_{1}\approx (\mu_{0}/\pi )\ln\left(D/g\right)$$
where $\mu_{0}$ and $\varepsilon_{0}$ are the vacuum permeability and permittivity, respectively, of $\varepsilon_{r}\;$, which is the relative permittivity of the medium in the vicinity, and the other parameters, *D*, *g*, *l*_1_, *l*_2_, *w*_2_, and *t*, are denoted in Figure 1a. As a result, by substituting Equations (1) and (2), the resonant frequency of single-layer metasurface is estimated as [54]
(3)$$f=\frac{1}{2\pi \sqrt{L_{1}C_{1}}}$$

Further, the series inductor *L*_s_ and shunt capacitor *C*_s_ in Figure 3 denote the dielectric substrate (or Rogers spacer) modeled by transmission line according to the Telegrapher’s equations. Specifically, inside the stop band determined by *L*_1_ and *C*_1_, the inductor of *L*_s_ and capacitor of *C*_s_ can be considered as short and open circuits, respectively, due to the relatively small values. Therefore, according to the *TL* equivalent circuits of two-port networks [54], the characteristic impedance (*Z_c_*), and its refection (*R*), and the transmission (*T*) coefficients in the stop band can be analytically approximated as
(4)$$Z_{c}=\frac{Z_{0}Z_{1}}{Z_{1}+Z_{0}}$$
and
(5)$$R=S_{11}=\frac{Z_{c}-Z_{0}}{Z_{c}+Z_{0}}$$
and
(6)$$T=S_{21}=\left(1+S_{11}\right)\frac{Z_{0}}{Z_{C}}$$
respectively, where $Z_{1}=R_{1}+1/j\omega C_{1}+j\omega L_{1}$ is the complex impedance of the series *LC* circuit. Although Equations (1)–(3) tentatively give the equivalent capacitance and the inductance and resonant frequencies, respectively, the lumped elements in Figure 3 can be further adjusted by fitting the resonant frequency, impedance *Z_c_*, and the scattering coefficients denoted by Equations (4)–(6) with the counterparts obtained by FEM calculations and impedance retrieval. As a result, spectral responses including the bandwidth and resonant frequency can be accurately modeled and interpreted.

### 2.2. Double Metasurfaces Cascaded with Intercoupling

Further, for a higher degree of freedom in spectral tuning, multiple metasurfaces cascaded in parallel with interlayer couplings are taken into investigations in the following subsection. First, a double-layer setup is constructed by cascading two parallel metasurfaces with one dielectric spacer of Rogers 3003 sandwiched in between, as shown in Figure 4a. To model its spectral behaviors, a lumped equivalent circuit is directly built by cascading two series LC circuits, as shown in Figure 4b.

Herein, the original series LC circuit is kept to model the top (first) metasurface that faces the incident electromagnetic wave, and the other LC circuit is introduced to model the bottom (second) metasurface. Herein, initial values of capacitances *C*_1_, *C*_2_ and inductances *L*_1_, *L*_2_ for both metasurfaces can be estimated individually with Equations (1) and (2). To model their interplay via the dielectric spacer, the same values of *L*_s_ and *C*_s_ are still used as those in Figure 3b. Further, a mutual inductance of *M* is introduced between two series LC branches. For a large interlayer gap, when both layers work in independent resonance modes and no mutual coupling exists (i.e., *M* = 0), the circuit in Figure 4b is simply a parallel combination of two series LC circuits and the spectral behaviors (resonance, transmission, etc.) can be readily obtained in a similar manner as it can in Equations (3)–(6). 

However, due to the sub-wavelength gap in our work, strong interlayer coupling exists between two metasurfaces, and a considerable mutual inductance, *M*, exists between two series LC circuits, as indicated in Figure 4b. In this situation, the total equivalent impedance of two series *LC* circuits in parallel with mutual inductance can be derived as
(7)$$Z=j\omega M+\frac{\left(Z_{1}-j\omega M\right)\left(Z_{2}-j\omega M\right)}{Z_{1}+Z_{2}-2j\omega M}=\frac{Z_{1}Z_{2}+\omega^{2}M^{2}}{Z_{1}+Z_{2}-2j\omega M}$$
where $Z_{1}=R_{1}+1/j\omega C_{1}+j\omega L_{1}$ and $Z_{2}=R_{2}+1/j\omega C_{2}+j\omega L_{2}$ denote the initial impedance of either series *LC* branch. The impedance *Z* in Equation (7) can be also regarded as the total equivalent impedance of the cascaded metasurfaces. 

Therefore, with the mutual inductance being considered, the resonant frequencies can now be found when impedance *Z* in Equation (7) approaches zero. For any fixed value of $M=k\sqrt{L_{1}L_{2}}$, either the positive (additive) or negative (subtractive) mutual inductances, there are constantly two zero points for *Z* = 0 in Equation (7), which indicate two separate frequencies for the band-stop behaviors of the cascaded metasurfaces. 

As a special case, when $C_{1}=C_{2}=C$, $L_{1}=L_{2}=L$ and *R*_1_ and *R*_2_ are infinitesimal and neglected, respectively, the initial two zero points converge to one. In this situation, either *LC* series branch has the same equivalent inductance, which is L+M for the additive mode or L−M for the subtractive mode of mutual inductance, respectively. Therefore, the equivalent circuit in Figure 4 works at two resonant frequencies of 1/($2\pi \sqrt{\left(L+M\right)C}$) and 1/($2\pi \sqrt{\left(L-M\right)C}$), which can be regarded as reversely split from 1/($2\pi \sqrt{LC}$) due to the different polarities of mutual inductance corresponding to different modes of coupling between two identical metasurfaces. Obviously, the stronger the coupling is, e.g., for a smaller interlayer gap, the larger the mutual inductance *M* is, and the other two resonant frequencies split farther away from each. In this manner, the stop-band bounded by two resonant frequencies that are tunable by adjusting the interlayer inductance (gap) becomes scalable due to the intercoupling between two metasurfaces. This principle also holds for the general case $C_{1}\neq C_{2}$, $L_{1}\neq \;L_{2}$ for two series *LC* branches cascaded in parallel with mutual coupling. Other than the mutual inductance *M* discussed above, the interlayer gap in between also leads to an extra coupling capacitance *C_m_*, which can be approximated as [53]
(8)$$C_{m}=0.2\varepsilon_{0}\varepsilon_{r1}gD/t_{1}$$
where $\varepsilon_{r1}$ and $h_{1}$ denote the permittivity and thickness of the dielectric spacer (Rogers 3003) filled in between, respectively. Herein, the interlayer capacitance Cm, which is proven to be much smaller than the intra-layer capacitance is [53], is equivalently incorporated into the single-layer capacitances $C_{1}$ and $C_{2}$ for simplicity. Therefore, the cascaded model works at distinctly split frequencies due to interlayer coupling, which enables the broadband scalable spectrum. The combined modes of electric resonance and interlayer coupling cause hybridized spectral behaviors of band expanding, as modeled by the *LC* circuit in Figure 4b. In addition, *L_s_* and *C_s_* also remain as short and open circuits within the stop band determined by *L*_1_, *C*_1_, *L*_2_, and *C*_2_. Therefore, given the total equivalent impedance in Equation (7), the spectral behaviors, including the resonant frequencies and the transmission coefficients, can be obtained similarly with Equations (4)–(6) and fitted to the FEM calculation results.

### 2.3. Triple Metasurfaces Cascaded with Intercoupling

Further, for higher degree of freedoms of spectral tuning for broadband regulation, a third metasurface is further introduced for a tri-layer cascaded model, and its equivalent circuit is built in a similar manner. As shown in Figure 5, another series LC circuit is introduced to model the third resonant mode of hybrid spectral behaviors, with intercoupling among each other. Herein, the *L*_1_ and *C*_1_ and *L*_2_ and *C*_2_ branches still model the top and middle layer metasurfaces, respectively, as above. The single-layer equivalent inductance *L*_3_ and capacitance *C*_3_ in the third branch can be estimated in a similar manner using Equations (1) and (2). Note that for large gaps between the adjacent layers of the triple-layer model, the cascaded model of three series LC circuits in Figure 5 still works with no mutual coupling (or inductances).

However, as it is similar to the double-layer case, strong coupling exists between the adjacent layers when the interlayer gaps reduce down to sub-wavelength level, and mutual inductances are also introduced among three series *LC* circuits, as shown in Figure 5b. For generality, *M*_1_ and *M*_2_ denote the mutual inductances due to top–middle and middle–bottom couplings, respectively. Still, two sets of series inductor *Ls*_1_, *Ls*_2_ and shunt capacitor *C_s_*_1_, C*s*_2_ are inserted in between to model the dielectric spacers inside the gap, as shown in Figure 5b. Similar to the double-layer model, the total equivalent impedance of three series *LC* circuits in parallel with mutual inductance can be derived as
(9)$$Z=\frac{\left(Z_{3}M_{1}{}^{2}+Z_{1}M_{2}{}^{2}\right)\omega^{2}+Z_{1}Z_{2}Z_{3}}{{\left(M_{1}-M_{2}\right)}^{2}\omega^{2}-2j\omega \left(M_{1}Z_{3}+M_{2}Z_{1}\right)+Z_{1}Z_{2}+Z_{1}Z_{3}+Z_{2}Z_{3}}$$
where *Z*_1_ and *Z*_2_ take the same expressions as those in Equation (7), and $Z_{3}=1/\left(\frac{1}{R_{3}+j\omega L_{3}}+j\omega C_{3}\right)$ denotes the impedance of the third series LC circuit. The resonant frequencies and transmission coefficients are obtainable according to Equations (4)–(6). Obviously, there are three zero points indicated in Equation (9), so three of them are resonant frequencies that constitute and shape the stop-band of the triple-layer cascaded model in this case. These frequencies can be regarded as the resonance modes of each *LC* branch hybridized with inter-coupling between the adjacent layers. By tuning each square ring length (*D*_1_, *D*_2_, or *D*_3_) or other geometries individually, the impedances *Z*_1_, *Z*_2_, or *Z*_3_ of each *LC* branch can be regulated in an independent manner according to Equations (1)–(3). Further, by changing the top–middle gap *h*_1_ or the middle–bottom gap *h*_2_, the mutual couplings or inductances are also adjustable, so are the resonance modes, as well as the spectral behaviors that are determined by the two adjacent layers. As a result, the whole spectrum, including the bandwidth, resonant frequencies, etc., turns out to be scalable in terms of the geometries of the individual layer and inter-coupling effect. Similar to the double-layer case, a smaller (or larger) gap between the adjacent layers for stronger (or weaker) mutual coupling in between cause two adjacent resonant frequencies to split farther (or closer) away from each other.

## 3. Results and Discussions

To verify our scheme for the scalable spectral regulation by cascaded metasurfaces with intercoupling, the lumped equivalent circuit models were analyzed and fitted to the numerical calculations by finite element method (FEM). First, the initial values of the lumped elements (capacitance, inductance, and resistance), characteristic impedances, and transmissions (or reflections) of the TL circuit models were calculated using Equations (1)–(9). Thereafter, accurate values were obtained by further fitting the analytical results to those obtained by FEM simulations. Herein, a retrieval procedure based on scattering coefficients was used to extract the characteristic impedance of a single metasurface according to the effective medium theory. Finally, the TL equivalent circuit models are used to instructively tune the spectral behaviors of single and cascaded metasurfaces, e.g., the resonant frequency, bandwidth, etc.

### 3.1. Numerical Verification of the Single-Layer Metasurface

First, for a single-layer metasurface working at the resonant frequency of 50 GHz, all the structural parameters were chosen to be: P = 2.4 mm, D = 1.44 mm, g = 0.12 mm, *w*_1_ = *w*_2_ = 0.12 mm, *l*_1_ = 0.5 mm, and *l*_2_ = 0.28 mm. The metal and dielectric (Rogers 3003) substrate thicknesses are 17.5 μm and 0.762 mm, respectively. After a fitting procedure following the FEM calculation, the lumped elements of the TL equivalent circuit model in Figure 3 were optimized as: *L*_1_ = 1.28 nH, C_1_ = 7.8 *fF*, R_1_ = 0.5 Ω, *L*_S_ = 0.7 nH, C_S_ = 7 *fF*, and Z_0_ = 377 Ω. As shown in Figure 6, the fitted transmission of the TL equivalent circuit model agrees well with the FEM simulation.

For spectral regulations through the structural geometries adjustment, the transmission spectra were calculated with varied lengths (D) of rectangular rings (denoted in Figure 1), as shown in Figure 7a. To model and interpret the spectral shifts caused by variations in the rectangular ring length, the spectra of TL circuit models were also fitted (Figure 7b) with the varied equivalent inductance *L*_1_, as indicated in Equation (2). Obviously, both the results fit well with the trend of spectral shifts, i.e., a larger ring length leads to a higher inductance and lower resonant frequencies. 

In a similar manner, the other parameters (e.g., *l*_1_, *l*_2_, and *g*) also subtly shift the resonant frequency of the single-layer metasurface as indicated in Equations (1) and (2). Upon the adjustment of each parameter, the equivalent impedance and transmission coefficient of the TL circuit models are obtained using Equations (1)–(6) and fitted to the FEM simulated results for accurate predictions. The numerical results of the FEM simulations and TL models reveal that the outer ring length *D* dominates over other parameters in shifting the resonance frequency for the single-layer metasurface.

### 3.2. Numerical Verification of Intercoupled Double Metasurfaces

For spectral regulation with more freedom, the cascaded models of multi-metasurfaces are explored for flexible tuning with more structural parameters, as mentioned in Section 2.2 and Section 2.3. First, the cascaded model of double-layer metasurfaces was established to work at two split resonant frequencies of around 55 GHz and 45 GHz. The top and bottom metasurfaces were used with the original spacer thickness of 0.762 mm and identical structural parameters of *P* = 2.0 mm, *D* = 1.4 mm, g = 0.12 mm, *w*_1_ = *w*_2_ = 0.12 mm, *l*_1_ = 0.48 mm, and *l*_2_ = 0.28 mm. Further, by fitting the ideal transmission spectrum of the TL model to that which was obtained by the FEM calculations (shown in Figure 8), all the lumped elements illustrated in Figure 4 were extracted and optimized as: *L*_1_ = 2.7 nH, C_1_ = 4.8 *fF*, R_1_ = 1 Ω, *L*_2_ = 2.7 nH, C_2_ = 2.85 *fF*, R_2_ = 1.5 Ω, *L*_S_ = 1.4 nH, and C_S_ = 3 *fF*.

In particular, different from the single-layer model, the broadband spectra of double metasurfaces are further scalable by varying the lengths (*D*_1_ or *D*_2_) of either of the square rings (top or bottom) or the gap in an independent manner. As predicted in Section 2.2, the cascaded model of the double metasurfaces operates at two split resonant frequencies that can be modeled by two inductively coupled *LC* circuits, as shown in Figure 8. As shown in Figure 9, the vector electric fields by FEM calculations also confirm the polarity of mutual inductance for two split resonant frequencies. Namely, the lower frequency mode operates with additive mutual inductance, while the higher frequency mode operates with subtractive mutual inductance, which agrees well with our predictions from the cascaded model of two identical metasurfaces.

Furthermore, as shown in Figure 10a, when the top or the bottom ring length increases to be 1600 μm or decreases to be 1200 μm, there is an obvious red or blue shift occurring to the transmission spectra, respectively. Agreeing well with our TL model in Figure 4, any ring length variations distinctly change the equivalent inductance, as well the resonant frequency of that layer or its equivalent *LC* branch. For the other layer without ring length variations, its frequency also shifts distinctly due to intercoupling or mutual inductance. Specifically, as shown in Figure 10a, when either of the square ring lengths (e.g., *D*_1_ for the top ring) decrease from 1400 μm to 1200 μm, the original higher frequency at ~55 GHz (e.g., for *D*_1_ = 1400) undergoes a distinct blue shift due to the reduction of its equivalent inductance (*L*_1_ for top ring) according to Equation (2). However, for the original lower resonant frequency at ~45 GHz, which is determined by an extra additive mutual inductance (beside the bottom layers *L*_2_ and *C*_2_), a reduction of either of the ring lengths (*D*_1_ or *D*_2_) to 1200 μm causes larger deviations between them, and meanwhile, reduced mutual inductance, *M*. As a result, the lower frequency of 1/($2\pi \sqrt{\left(L+M\right)C}$) is also increased, which is in good agreement with the discussions in Section 2.2. 

Apart from the ring length (*D*_1_ and *D*_2_) tuning, the interlayer gap or dielectric spacer (*h*_1_) in between also determines the intercoupling for direct spectral regulations. As shown in Figure 10 (red and blue dashed curves), when the gap shrinks from 762 μm to 506 μm, an obviously blue shift occurs at the original lower frequency, while a red shift occurs at a higher frequency. In good agreement with the predictions of the TL model about mutual inductance discussed in Section 2.2, the larger gap causes two modes to split farther away from each other because the modes with lower and higher frequencies are determined by additive and subtractive mutual inductances, respectively. Namely, the smaller gap causes stronger interlayer coupling and larger deviations in the equivalent inductances of two *LC* branches in Figure 4. On contrary, as shown in Figure 10 (black curve), when the gap increases to be a very large value (3800 μm), two resonant modes approach each other, and ultimately, merge into one mode because the mutual inductance of two series *LC* branches or the inter-coupling of two metasurfaces vanishes. 

In a similar manner, other structural geometries of *l*_1_, *l*_2_, and *g* in both metasurfaces are also tunable for spectral regulations via the double-layer cascaded model. However, we omitted the discussions for such parameters here for simplicity. As a result, by varying the geometries of the top and bottom rings, as well as the interlayer gap, the resonant frequencies and bandwidth are scalable in both a global and local manner.

### 3.3. Numerical Verification of Intercoupled Triple Metasurfaces

For higher freedoms of scalable spectral tuning, the triple-layer cascaded model was verified in a similar manner. As an example, the model was established to work at three split resonant frequencies of 55 GHz, 45 GHz, and 40 GHz, as shown in Figure 11. In this case, the three metasurfaces are customized with slightly different geometries for optimized spectral tuning. For the top metasurface, the parameters of *P*_1_ = 2.4 mm, *D*_1_ = 1.4 mm, *g* = 0.14 mm, *w*_1_ = 0.16 mm, *w*_2_ = 0.1 mm, *l*_1_ = 0.48 mm, and *l*_2_ = 0.32 mm are used, as denoted in Figure 1. For the middle one, the parameters of *P*_2_ = 2.4 mm, *D*_2_ = 1.3 mm, *g* = 0.16 mm, *w*_1_ = *w*_2_ = 0.16 mm, *l*_1_ = 0.4 mm, and *l*_2_ = 0.28 mm are used. For the bottom metasurface, the parameters of *P*_3_ = 2.4 mm, *D*_3_ = 1.63 mm, *g* = 0.16 mm, *w*_1_ = *w*_2_ = 0.16 mm, *l*_1_ = 0.56 mm, and *l*_2_ = 0.32 mm are used. By estimating the initial values of all the lumped equivalent elements based on Equations (1) and (2), the TL equivalent circuit model is established and perfectly fitted to the FEM results, as shown in Figure 11. Via the fitting procedure, accurate values of the lumped elements denoted in Figure 5 are determined as *L*_1_ = 3 nH, C_1_ = 5.75 *fF*, R_1_ = 2.5 Ω, *L*_2_ = 2.83 nH, C_2_ = 3.9 *fF*, R_2_ = 0.9 Ω, *L*_3_ = 3.22 nH, C_3_ = 2.15 *fF*, R_3_ = 6.5 Ω, *L*_S1_ = 1.2 nH, C_S1_ = 2.1 *fF*, *L*_S2_ = 1.8 nH, and C_S2_ = 2.1 *fF*.

As indicated in Section 2.3, the resonant modes at ~40 GHz, ~45 Hz, and ~55 GHz in Figure 11 are accordingly modeled by the *L*_1_, *C*_1_ branch, the *L*_2_, *C*_2_ branch, and the *L*_3_, *C*_3_ branch together with mutual inductances between the adjacent branches, as shown in Figure 5. As indicated by the tri-layers cascading model, the highest frequency ~55 GHz is determined by the middle layer intercoupling with the top layer; the middle frequency ~45 GHz is determined by top layer intercoupling with the middle layer; the lowest frequency ~40 GHz is determined by bottom layer hybridized with the intercoupling with the middle layer. 

Further, as for the scalable spectral tuning by intercoupling, three resonant modes with different frequencies are affected by coupling between the top and middle layers or that between the middle and bottom layers in a different manner. As shown in Figure 12a, at the lowest frequency of ~40 GHz, the FEM-calculated vector electric fields show additive mutual inductances (M2 and M1) for both top–middle coupling (anti-phase) and middle–bottom coupling (anti-phase). At the middle frequency of ~45 GHz, the top–middle coupling (anti-phase) dominates with an additive mutual inductance (M2) and the middle–bottom coupling (in-phase), which is much weaker than top–middle coupling is, shows a subtractive mutual inductance (M1), as confirmed by the vector electric fields in Figure 12b. At the highest frequency of ~55 GHz, both the top–middle and middle–bottom couplings (both in-phase ones) exhibit subtractive mutual inductances, as shown in Figure 12c. In either case, the middle–bottom coupling is much weaker than the top–middle coupling is, which means the mutual inductance of M2 dominates over M1 (also confirmed by our TL model fitting results). 

Therefore, other than the individual *LC* branch, the highest frequency (~55 GHz), middle frequency (~45 GHz), and lowest frequency (~40 GHz) are coupled with the subtractive mutual inductance of *M*_2_, the additive mutual inductance of *M*_2_ (subtractive mode of *M*_1_ is weak and neglected), and the additive inductance of *M*_1_, respectively. 

First, the top–middle coupling is used for the scalable tuning of the highest and middle frequencies. As confirmed by the FEM results in Figure 13a, reducing (increasing) the dielectric (Rogers 3003) spacer thickness *h*_1_ between the top and middle layers causes a distinct red (blue) shift to the middle frequency (~45 GHz) and a blue (red) shift to the largest frequency (~55 GHz), i.e., it drives the two frequencies farther away from (closer to) each other, due to the stronger (weaker) coupling for the larger (smaller) mutual inductance *M*_2_. Moreover, the middle–bottom coupling is also used to tune the lowest frequency (~40 GHz) via the additive mutual inductance (*M*_1_), since the middle frequency (45 GHz) is mainly determined by the additive mutual inductance of *M*_2_. As also confirmed by FEM results in Figure 13b, changing the dielectric gap of *h*_2_ for middle–bottom coupling causes distinct red shifts to the lowest frequency (~40 GHz), but with a minor impact on other two frequencies.

Similar as the double-layer model, the spectral behaviors of the triple-layer model are also tunable by changing the square ring length, *D*_1_, *D*_2_, and *D*_3_, of three layers individually. As shown in Figure 14a, increasing (decreasing) the top ring length, *D*_1_, from 1400 μm to 1600 μm (1200 μm) causes a distinct red (blue) shift to both the largest frequency (~55 GHz) and middle frequency (~45 GHz), but with almost no impact on the lowest frequency (~40 GHz). This agrees well with our TL model above because the top ring length (*D*_1_) determines the middle and largest frequencies via the equivalent inductance of *L*_2_ (or *L*_3_) and the mutual inductance (*M*_2_) between *L*_2_ (top layer) and *L*_3_ (middle layer). Namely, when a larger *D*_1_ induces a larger *L*_2_ according to Equation (2) for a reduced middle frequency, it also leads to weaker intercoupling with the middle layer (due to farther away from *D*_2_), and thus, a smaller subtractive mutual inductance of *M*_2_ to *L*_3_, which ultimately causes a reduction (or red shift) to the highest frequency, and vice versa for the smaller ring length of *D*_1_.

Further, increasing (decreasing) the middle ring length *D*_2_ from 1300 μm to 1500 μm (1100 μm) reduces (increases) the lowest frequency (~40 GHz) distinctly, as well as the other two frequencies, as shown in Figure 14b. Similar to the results in Figure 13a, the middle layer determines the largest and middle frequencies via the top–middle couplings in terms of *M*_2_ and the equivalent inductance of *L*_2_ (or *L*_3_). However, the difference here is the lowest frequency that is also tunable by the middle layer in terms of middle–bottom coupling (*M*_1_). As an example, when *D*_2_ is reduced to 1100 μm, the inductance of *L*_3_ decreases to blue shift the largest frequency, and the mutual inductances *M*_2_ (top–middle coupling) and *M*_1_ (middle–bottom coupling), both in additive modes as indicated in above discussions, also decrease to cause a blue shift to the middle and lowest frequencies, due to larger differences between *D*_2_ and *D*_1_ or *D*_2_ and *D*_3_. 

Finally, for the bottom metasurface, increasing the ring length of *D*_3_ causes a more distinct red shift to the lowest frequency (~40 GHz), with a relatively minor impact on the two higher frequencies, as shown in Figure 14c. As also agrees well with our TL models, since the bottom ring length *D*_3_ mainly involves middle–bottom coupling, which is unable to have a minor impact on the other frequencies. However, when we were reducing the ring length of *D*_3_ to 1430 μm (much closer to *D*_2_ and *D*_1_), the middle–bottom coupling increases significantly, and the subtractive mutual inductance of *M*_1_ to causes a distinct red shift to the middle frequency, but with almost no impact on the highest frequency.

### 3.4. Experimental Verifications

As discussed above, by optimally changing the structural geometries of individual layers of cascaded metasurfaces and taking full advantages of interlayer coupling in between them, hybridized resonant modes with split frequencies can be leveraged and tuned in an independent manner for optimal spectral scaling. Therefore, to further experimentally verify the feasibility of the cascaded model for spectral regulations, the optimized devices of both the double- and triple-cascaded metasurfaces were fabricated by using the optimized structural parameters as those obtained in the beginning of Section 3.2 and Section 3.3. Herein, Rogers 3003 is embedded as the dielectric spacer in between the cascaded metasurfaces. After sample design and fabrication, the transmission spectra were then characterized in an MM wave experimental testing platform that comprises a pair of horn antennas for MM wave transmitting and receiving and an Agilent vector network analyzer for MM signal analysis. As already shown previously in Figure 8 and Figure 11, the FEM calculations and TL model agree well with each other. Or, in other words, our TL models accurately interpret the spectral behaviors of the cascaded metasurfaces. Therefore, by direct experimental testing of the spectra of two devices that are already designed and optimized according to the TL models, the validities of our TL model for spectral analysis and prediction can be confirmed directly by fitting the measured spectra to the FEM calculated results. 

As a result, as shown in Figure 15a,b, the experimentally measured transmission spectra agree well with the FEM calculations (also shown in Figure 8 and Figure 11) for both the double-layered and triple-layered devices. The insets in Figure 15a,b denote the microscopic images of the fabricated device samples, respectively. Obviously, with optimal and scalable spectral tuning, the double- and triple-layer models lead to substantially expanded transmissive stop bands compared with that of the single metasurface in Figure 6. Both the experimental results of double- and triple-layer devices show perfect agreement with the FEM calculations and TL models, which also confirm the validity of our scheme for spectral tuning by simply cascading multiple metasurfaces with similar or the same geometries.

## 4. Conclusions

We have demonstrated broadband spectral regulation with flexible scalability by leveraging the intercoupling effects among cascaded metasurfaces in the MMs regime. Double- and triple-layer cascaded models are constructed by composite meta-atoms composed of Jerusalem resonant crosses and square rings with band-stop behaviors. The resonance modes of periodic composite atoms hybridized with interlayer couplings effects are interpreted and modeled in terms of the simplified TL equivalent circuits to guide the scalable tuning of broad spectral behaviors. 

As a proof of concept, the double-layer and triple-layer cascaded devices are constructed and confirmed to scale the narrow stop-band of around 50 GHz for a single-layer metasurface up to a broadened range between 40 GHz and 55 GHz, with an ideal sideband. The experimentally measured results show nicely scaled and enlarged stop-band spectra, which agree well with the predictions of our TL models and FEM calculations. As a result, our methodology based on hybridized resonances with interlayer coupling by cascaded metasurfaces potentially provide an exemplary approach for spectral tuning with broadband scalability in MMs, THz, and related applications.

## Figures and Tables

**Figure 1 materials-16-02013-f001:**
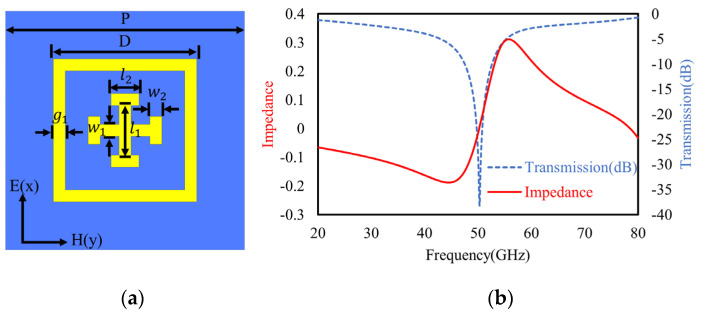
(**a**) The layout of one unit of single-layer metasurface; (**b**) the band-stop transmission spectrum and the retrieved imaginary wave impedance at TEM wave incidence. Structural parameters are illustrated in inset (**a**).

**Figure 2 materials-16-02013-f002:**
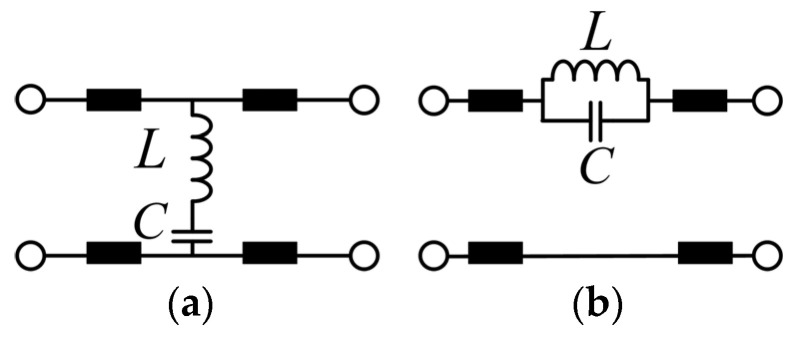
Two basic types of lumped equivalent circuits for the single-layer metasurface shown in Figure 1; (**a**) TL circuit A: series LC circuit; (**b**) TL circuit B: parallel LC circuit.

**Figure 3 materials-16-02013-f003:**
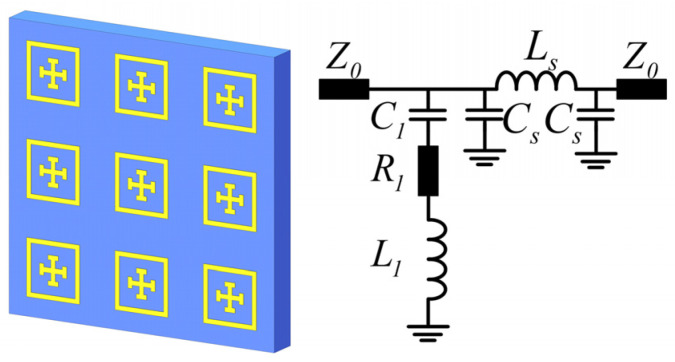
(**a**) The single-layer metasurface and (**b**) its equivalent circuit by considering the equivalent lumped capacitance, inductance, and resistance of the composite atoms of metal rings and Jerusalem crosses.

**Figure 4 materials-16-02013-f004:**
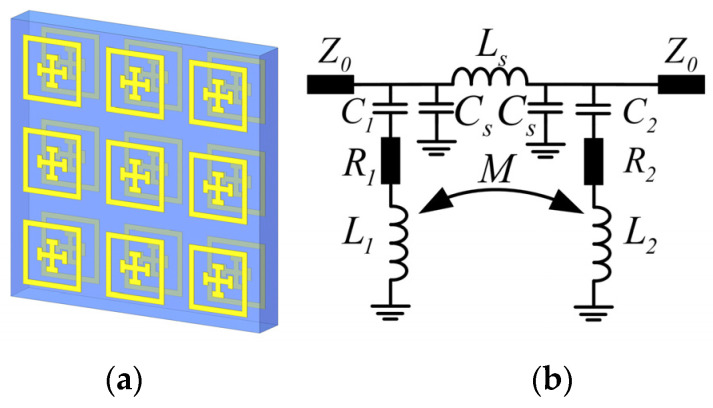
(**a**) The double-layer cascaded metasurfaces and (**b**) its equivalent lumped circuit model via two cascaded series LC circuits.

**Figure 5 materials-16-02013-f005:**
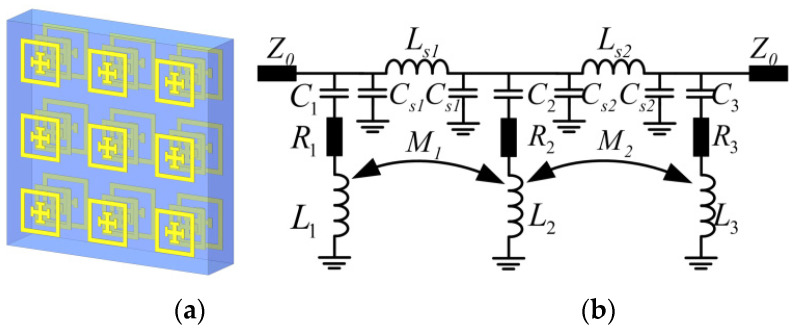
(**a**) The triple-layers cascaded metasurfaces and (**b**) the simplified TL equivalent circuit with one series LC circuit shunted plus two parallel LC circuits to mode the hybrid resonance modes with interlayer couplings.

**Figure 6 materials-16-02013-f006:**
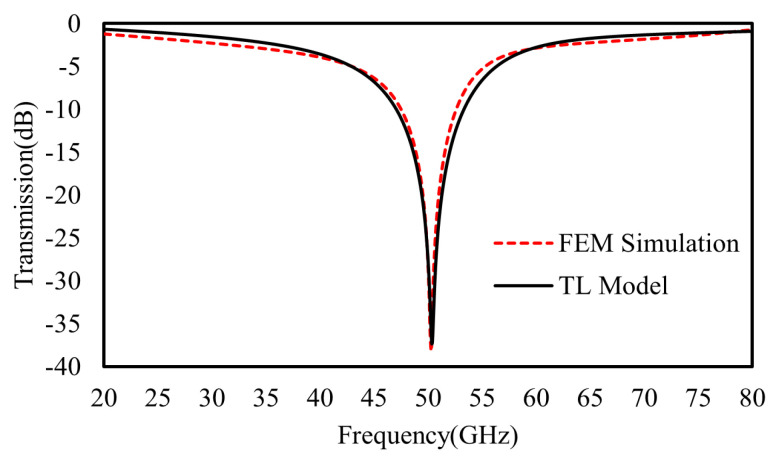
The FEM calculated (red dash) transmission coefficient of single metasurface and the fitted one (black) by TL model in Figure 3.

**Figure 7 materials-16-02013-f007:**
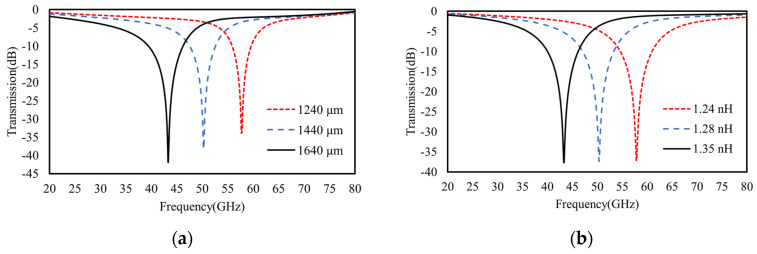
(**a**) The FEM calculated transmission spectra of single-layer metasurfaces for varied lengths (D) of rectangular rings in comparison to (**b**) the fitted transmission spectra of the TL equivalent circuit models with varied inductance *L*_1_, as denoted in Equation (2), in accordance with the varied lengths (D) of rectangular rings in (**a**).

**Figure 8 materials-16-02013-f008:**
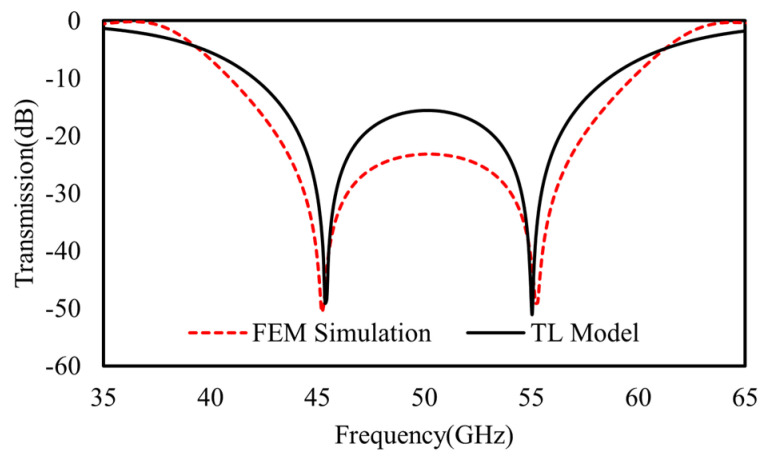
The transmissions spectrum of double-layers cascaded metasurface by FEM simulation (red dash) and the fitted spectrum (black dot) of ideal TL equivalent circuit model in Figure 4.

**Figure 9 materials-16-02013-f009:**
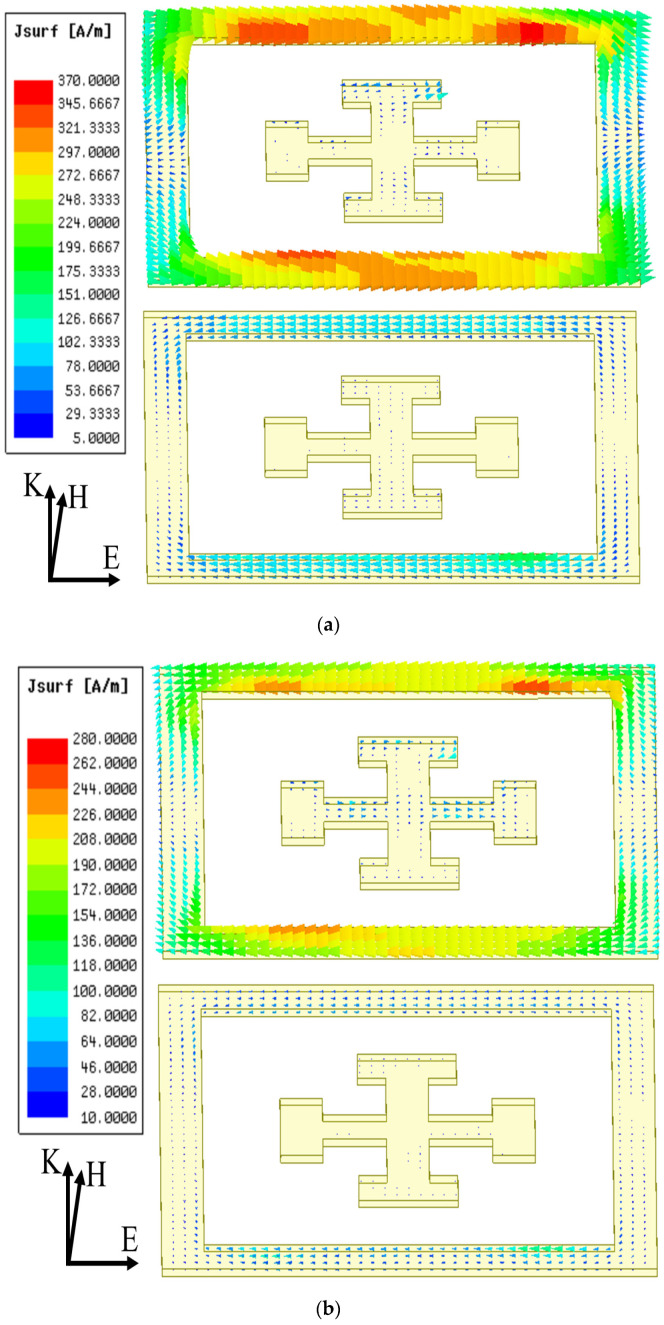
The FEM calculated electric field distributions show (**a**) the additive mutual inductance (anti-phase resonances at both layers) at 45 GHz and (**b**) subtractive mutual inductance (in-phase resonances at both layers) at 55 GHz.

**Figure 10 materials-16-02013-f010:**
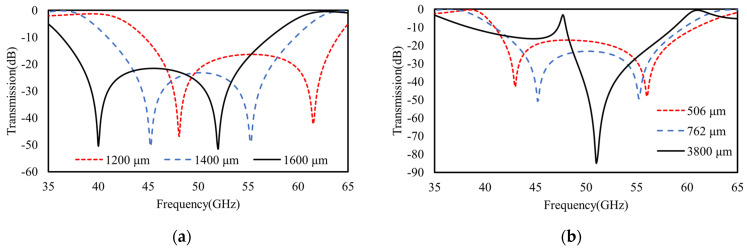
The FEM calculated transmission spectra of double-layer cascaded model. (**a**) The broadband scalable behaviors achieved by varying the ring length *D*_1_ of the top metasurface or *D*_2_ of bottom metasurface from 1400 μm to 1200 μm and 1600 μm and (**b**) by varying the interlayer gap (or spacer width *h*_1_) between the top and bottom metasurfaces.

**Figure 11 materials-16-02013-f011:**
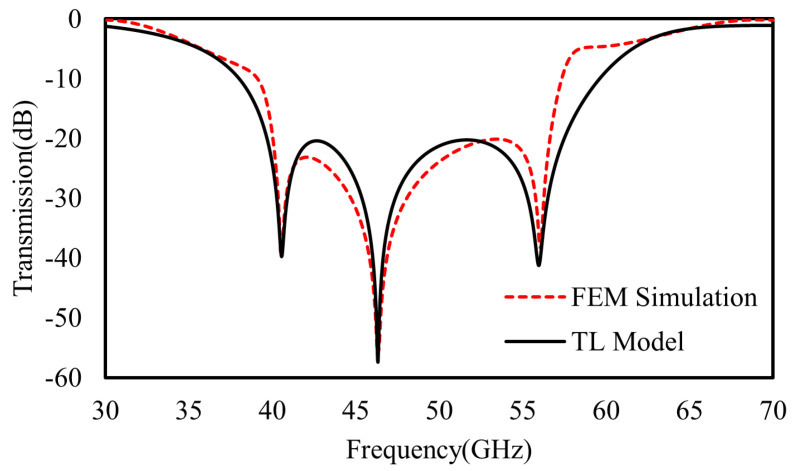
The FEM calculated transmission coefficient (red dash) of tri-layers cascaded model and the fitted one (black dot) by TL model in Figure 5.

**Figure 12 materials-16-02013-f012:**
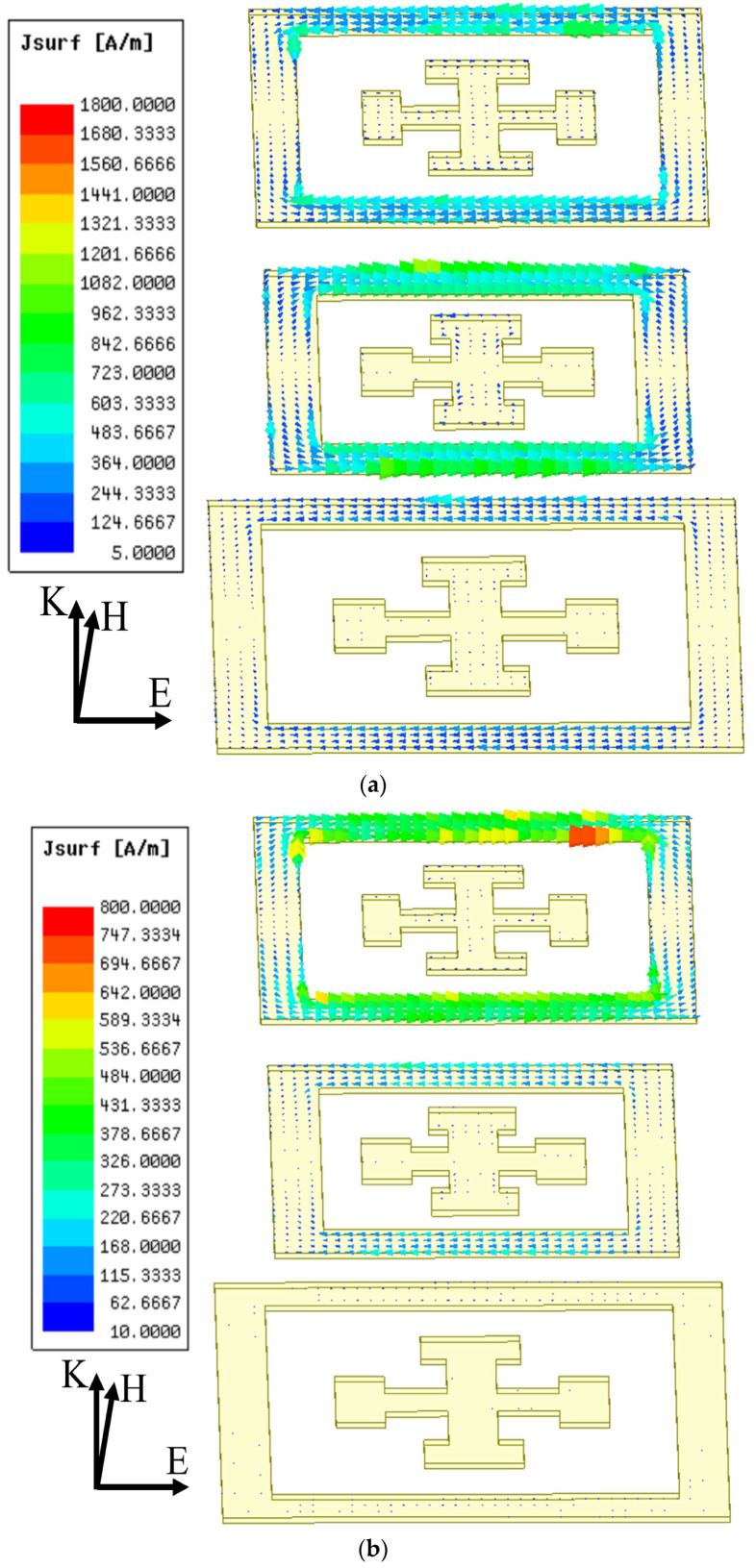
The FEM calculated electric field distributions at (**a**) the lowest frequency (40 GHz), (**b**) the middle frequency (45 GHz), and (**c**) the highest frequency (55 GHz). Obviously, the middle–bottom coupling is much weaker than the top–middle coupling is.

**Figure 13 materials-16-02013-f013:**
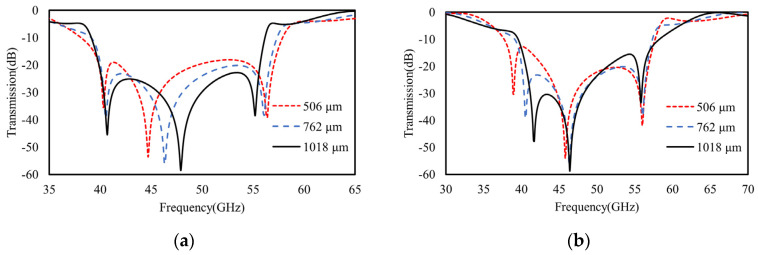
The FEM calculated transmission spectra of tri-layers model that is scalable by varying (**a**) the thickness of top–middle coupling dielectric spacer and (**b**) the thickness of the middle–bottom coupling dielectric spacer.

**Figure 14 materials-16-02013-f014:**
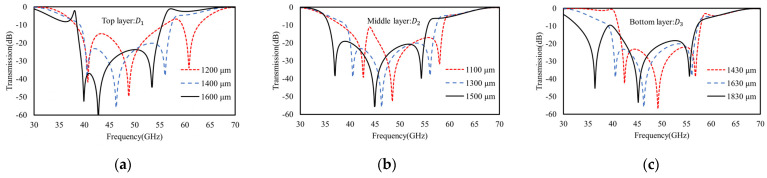
The FEM calculated spectra of tri-layers model that is scalable by varying (**a**) the square ring length *D*_1_ of top metasurface, (**b**) the square ring length *D*_2_ of middle metasurface, and (**c**) the square ring length *D*_3_ of bottom metasurface.

**Figure 15 materials-16-02013-f015:**
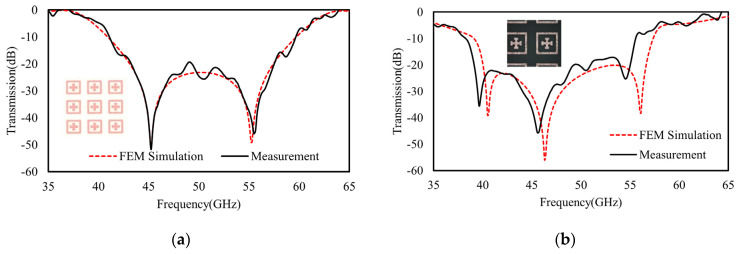
The measured transmission spectra of (**a**) double-layer cascaded model and (**b**) triple-layer cascaded model in comparison with their FEM calculated results. The inset shows the meta-atoms geometries captured using an optical microscope.

## Data Availability

Data is contained within the article.

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
