# Peer review of "Intercoupling of Cascaded Metasurfaces for Broadband Spectral Scalability"

_materials, 2023, doi:10.3390/ma16052013_

Round 1

Reviewer 1 Report

Manuscript no: 2220918

This is review report on “Intercoupling of cascaded metasurfaces for scalable broadband …. ” submitted by Zhou et al. This work deals with realizing broadband metamaterials. Broadband MM is important direction, particularly considering upcoming THz/6G communications, that’s appreciable. This work started started with optimizing the designs using TL and numerical methods, that makes sense. However, I feel the manuscript needs lot of improvements before acceptance.

1)      English seems to be fine, don’t see major problem

2)      Figures really require improvements. Particularly, the surface current plots, figure 9, figure 12. It looks like these figures are pressed by hands, does not look like journal picture. Also more clarity require, increase in font size etc.

3)      Primarily the demonstrated MM works through overlapping of different resonance modes. This concept is not new. In a recent work, similar broadband MM is shown, please refer to, Plasmonics, 15, 1925 (2020). Its surprising to see there is no mention of this work. However, too many old works from very established groups are cited. I feel refrences must be improved.

4)      I do not understand the term “scalable”. MMs are indeed scalable, that’s one of the important feature of these artificial materials, hence popular across the electromagnetic spectrum. What exactly authors meant to say by “scalable”. It needs to be specified clearly. Otherwise better to drop from the title atleast. Looks redundant.

Overall this work is simple and reasonably straight forward. I don’t see any major technical glitch. But presentations must be improved. Recommending major revision.

Author Response

This is review report on “Intercoupling of cascaded metasurfaces for scalable broadband …. ” submitted by Zhou et al. This work deals with realizing broadband metamaterials. Broadband MM is important direction, particularly considering upcoming THz/6G communications, that’s appreciable. This work started started with optimizing the designs using TL and numerical methods, that makes sense. However, I feel the manuscript needs lot of improvements before acceptance.

1)      English seems to be fine, don’t see major problem

2)      Figures really require improvements. Particularly, the surface current plots, figure 9, figure 12. It looks like these figures are pressed by hands, does not look like journal picture. Also more clarity require, increase in font size etc.

Re: Yes, we have changed those figures back to the ones with original size and better definition. Thanks for pointing out that.

3)      Primarily the demonstrated MM works through overlapping of different resonance modes. This concept is not new. In a recent work, similar broadband MM is shown, please refer to, Plasmonics, 15, 1925 (2020). It’s surprising to see there is no mention of this work. However, too many old works from very established groups are cited. I feel references must be improved.

Re: Yes, thanks for your good advices, we’ updated our references by adding your suggested one and other suitable ones. The suggested one belongs to one type of methods using composite cells for broadband tuning, which we have also discussed in the paper. Per your suggestions, we also added more recent references that treat interactions or overlapping of resonance modes by multi-layers, mainly in the 2nd paragraph of the introduction part.

4)      I do not understand the term “scalable”. MMs are indeed scalable, that’s one of the important feature of these artificial materials, hence popular across the electromagnetic spectrum. What exactly authors meant to say by “scalable”. It needs to be specified clearly. Otherwise better to drop from the title at least. Looks redundant.

 Re: Yes, agree. Thanks for your good advices. We hope to express the “broadband” and “scalable” spectral control by our method. To make it more clearly, we drop “scalable” and “regulations” and use scalability in the end.

Reviewer 2 Report

This paper presents a combination of theoretical and experimental studies of composite metamaterials in the millimeter wave range (50 Ghz).

The authors model a single layer, than introduce interlayer coupling and demonstrate that this approach is fruitful.

The topic is rather simple yet very interesting for the field of metamaterials. The results are scientifically sound and fit well this Special Issue.

Please find my comments/questions below:

1. Experimental details must be presented more clearly and thoroughly.

2. The authors should present an overview of the perspective for the further development. What is the goal of this research? Narrow bandpass filters? Anti-reflective coatings? 

3. Why Jerusalem cross-shaped elements were chosen? Is it due to simplicity?

4. Appendix A and Appendix B must be removed

Author Response

This paper presents a combination of theoretical and experimental studies of composite metamaterials in the millimeter wave range (50 Ghz).

The authors model a single layer, than introduce interlayer coupling and demonstrate that this approach is fruitful.

The topic is rather simple yet very interesting for the field of metamaterials. The results are scientifically sound and fit well this Special Issue.

Please find my comments/questions below:

  1. Experimental details must be presented more clearly and thoroughly.

Re: Yes, revised. Thanks for your kindly reminding. We have proof-checked the numerical parts and revised the experimental part.

  1. The authors should present an overview of the perspective for the further development. What is the goal of this research? Narrow bandpass filters? Anti-reflective coatings? 

Re: Yes. Thanks for your nice reminder. We added a few statements to our work at the end of introduction part, i. e. to provide an approach for spectral tuning with broadband scalability in MMs, THz and related applications.

  1. Why Jerusalem cross-shaped elements were chosen? Is it due to simplicity?

Re: Yes, Jerusalem cross-shaped elements is simple for analysis and we focus on theoretical analysis rather than structure innovation.

  1. Appendix A and Appendix B must be removed

Re: Yes. they are removed. Thank you again for your time to review our manuscript.

Round 2

Reviewer 1 Report

Current version can be accepted for publication.